# Lansoprazole protects hepatic cells against cisplatin-induced oxidative stress through the p38 MAPK/ARE/Nrf2 pathway

**Naoko Yamagishi**[ORCID]**\*, Yuta Yamamoto, Toshio Nishi, Takao Ito, Yoshimitsu Kanai**

Department of Anatomy and Cell Biology, Graduate School of Medicine, Wakayama Medical University, Wakayama, Japan

\* ymg-n@wakayama-med.ac.jp

**Data Availability Statement:** All relevant data are within the manuscript and its Supporting information files.

## Abstract

Lansoprazole, a proton pump inhibitor, can exert antioxidant effects through the induction of the nuclear factor erythroid 2-related factor 2 (Nrf2) pathway, independently of the inhibition of acid secretion in the gastrointestinal tract. Lansoprazole has been reported to provide hepatoprotection in a drug-induced hepatitis animal model through the Nrf2/heme oxygenase-1 (HO1) pathway. We sought to investigate the molecular mechanism of cytoprotection by lansoprazole. An *in vitro* experimental model was conducted using cultured rat hepatic cells treated with lansoprazole to analyze the expression levels of Nrf2 and its downstream genes, the activity of Nrf2 using luciferase reporter assays, cisplatin-induced cytotoxicity, and signaling pathways involved in Nrf2 activation. Lansoprazole treatment of rat liver epithelial RL34 cells induced transactivation of Nrf2 and the expression of the Nrf2-dependent antioxidant genes encoding HO1, NAD(P)H quinone oxidoreductase-1, and glutathione S-transferase A2. Furthermore, cycloheximide chase experiments revealed that lansoprazole prolongs the half-life of the Nrf2 protein. Notably, cell viability was significantly increased by lansoprazole treatment in a cisplatin-induced cytotoxicity model. Moreover, the siRNA knockdown of Nrf2 fully abolished the cytoprotective effect of lansoprazole, whereas the inhibition of HO1 by tin-mesoporphyrin only partially abolished this. Finally, lansoprazole promoted the phosphorylation of p38 mitogen-activated protein kinase (MAPK) but not that of the extracellular signal-regulated kinase or the c-Jun N-terminal kinase. Using SB203580, a specific inhibitor for p38 MAPK, the lansoprazole-induced Nrf2/antioxidant response elements pathway activation and cytoprotective effects were shown to be exclusively p38 MAPK dependent. Lansoprazole was shown by these results to exert a cytoprotective effect on liver epithelial cells against the cisplatin-induced cytotoxicity through the p38 MAPK signaling pathway. This could have potential applications for the prevention and treatment of oxidative injury in the liver.

## Introduction

Nuclear factor-erythroid 2-related factor 2 (Nrf2) plays a crucial role in the transcriptional regulation of antioxidant and detoxifying genes in various cell types, including hepatocytes [1, 2]. The activation of the Nrf2 pathway prevents liver injury caused by many hepatotoxicants [3,

**Funding:** This work was supported in part by a Japan Society for the Promotion of Science (JSPS) KAKENHI grant (grant no. 17K15963) to Naoko Yamagishi. The funders had no role in study design, data collection and analysis, decision to publish, or preparation of the manuscript.

**Competing interests:** The authors have declared that no competing interests exist.

4]. Consequently, Nrf2 has recently been implicated as a new therapeutic target for the treatment of liver diseases. Under normal conditions, Nrf2 binds to a repressor Kelch-like ECH-associated protein 1 (Keap1) in the cytoplasm as an inactive complex. Keap1 functions as a substrate adaptor protein for a Cullin3-dependent E3 ubiquitin ligase complex and induces a rapid proteasomal degradation of Nrf2 [5]. Keap1 contains several reactive cysteine residues that function as sensors for cellular oxidative stress [6–8]. Modification of these critical cysteine residues of Keap1 by reactive oxygen species (ROS) inactivates the E3 ubiquitin ligase and stabilizes Nrf2, which subsequently accumulates and translocates into the nucleus. Nuclear Nrf2 binds to antioxidant response elements (AREs) that are located in the promoter region of genes and encode various antioxidant and phase 2 detoxifying enzymes such as heme oxygenase-1 (HO1), NAD(P)H quinone oxidoreductase-1 (NQO1), and glutathione S-transferase A2 (GSTA2) [9–11]. The ARE-mediated upregulation of these enzymes facilitates the removal of toxic agents and ROS, thereby providing a protection against liver injury.

Exposure of cells to Nrf2 inducers, like oxidative stress stimuli, activates Nrf2/ARE-mediated response through different intracellular signaling pathways. Mitogen-activated protein kinase (MAPK) pathways are known to regulate Nrf2/ARE-driven gene expression [12–14]. MAPKs are serine/threonine protein kinase that play a central role in the signaling cascade regulating cellular processes, such as cell proliferation, differentiation, and apoptosis. Three major MAPK subfamilies have been extensively studied: extracellular signal-regulated kinases (ERKs), c-Jun N-terminal kinases (JNKs), and p38 MAPK. Each kinase establishes, in principle, parallel and independent signaling pathways. Depending on the cellular and stimulatory context, there is often significant cross-talk between each of the kinases because they can respond to common upstream activators and phosphorylate common downstream targets. Alam *et al.* reported that cadmium-induced HO1 gene expression requires the sequential activation of the p38 MAPK pathway and Nrf2 in human breast cancer MCF-7 cells [12]. In contrast, in human hepatoblastoma HepG2 cells, JNK activated the induction of Nrf2/ARE-mediated gene expression by the overexpression of common upstream kinases, whereas p38 MAPK showed the opposite effects [13]. Additionally, exposure of HepG2 cells to the chemical pyrrolidine dithiocarbamate, the activation of both ERK1/2 and p38 MAPK are required for induction of the ARE-mediated γ-glutamylcysteine synthetase subunit genes [14]. Based on these observations, the capability of lansoprazole to upregulate expression of antioxidant genes via the MAPK/ARE/Nrf2 pathway was investigated using rat hepatic cells.

Cisplatin is a chemotherapeutic agent commonly used to treat various solid tumors. Cisplatin-induced cytotoxicity has been described as a function of DNA crosslinking, followed by formation of DNA lesions. Cisplatin has been recently shown to significantly increase the generation of ROS that can cause organ toxicity, including hepatotoxicity [15–18]. Oxidative stress is the one of the most important mechanisms involved in cisplatin toxicity.

Lansoprazole is a potent proton pump inhibitor that is often used to treat acid-related disorders, including gastric and duodenal ulcers, gastro-esophageal reflux disease, and nonulcer dyspepsia. It strongly reduces the secretion of gastric acid by blocking $H^+/K^+$-ATPase in gastric parietal cells. In addition, it exerts anti-inflammatory and antioxidant effects against necrotizing agent-induced gastric lesions through a mechanism that acts independently of inhibition of acid secretion [19–22]. Several studies have demonstrated a lansoprazole-induced expression of the antioxidant protein HO1 in gastric epithelial cells [23, 24], endothelial cells [25], and neutrophils [26] as well as in the small intestines of rats [27–29], thereby protecting these cells from oxidative stress. HO1 is a stress-inducible protein, and its reaction products from heme degradation have been linked to cytoprotection. Takagi *et al.* proposed that the induction of HO1 by lansoprazole can occur by Nrf2 activation in rat gastric mucosal cells [23]. The induction of HO1 both in the gastrointestinal mucosa and in the liver can have

antioxidant and anti-inflammatory properties [30, 31]. We previously reported that lansoprazole attenuates the drug-induced oxidative liver injury in rats through the Nrf2/HO1 pathway [32]. Furthermore, lansoprazole treatment can suppress the expression of proinflammatory cytokines and the progression of hepatic fibrogenesis induced by a choline-deficient L-amino acid-defined diet [33]. However, how lansoprazole activates the Nrf2 pathway in hepatic cells remains unclear. Here, we focus on the role of the Nrf2 pathway to provide further insight into our previous findings on the molecular mechanisms of lansoprazole-mediated hepatoprotection. The present study comprises an *in vitro* experimental model that simulates the cisplatin-induced oxidative stress in rat hepatic cells.

## Materials and methods

### Chemicals and reagents

Lansoprazole was purchased from the Tokyo Chemical Industry Co. (Tokyo, Japan). p38 MAPK inhibitor SB203580 and protein biosynthesis inhibitor cycloheximide (CHX) were purchased from the FUJIFILM Wako Pure Chemical Co. (Osaka, Japan). Cisplatin was purchased from Nichi-Iko Pharmaceutical Co. (Toyama, Japan). The specific HO1 inhibitor tin-meso-porphyrin IX (SnMP) was purchased from BIOMOL Research Inc. (Plymouth Meeting, PA, USA).

### Cell culturing, transfection, and treatment

The untransformed rat hepatic cell line RL34 was obtained from the Japanese Collection of Research Bioresources Bank (JCRB0247). Cells were maintained in Dulbecco's modified Eagle medium (DMEM) without phenol red (FUJIFILM Wako Pure Chemical Co., Osaka, Japan) and supplemented with 10% (v/v) heat-inactivated fetal bovine serum (FBS) (Sigma-Aldrich, St. Louis, MO, USA), at 37°C with 5% $CO_2$. Plasmid transfection was performed using jet-PRIME transfection regent (Polyplus-transfection S.A., Illkirch, France). Lansoprazole was dissolved in dimethyl sulfoxide (DMSO) (FUJIFILM Wako Pure Chemical Co.). Cells were treated with 100 μM of lansoprazole; as a vehicle control, an identical amount of DMSO was added to the media.

### Cell viability assay

The cell viability assay was performed by using the Cell Titer 96 Aqueous One Solution Cell Proliferation Assay (Promega Corp., Madison, WI, USA). Cells were plated in a 96-well flat-bottom plate and cultured in 100 μL of DMEM containing 10% FBS. Cells were pretreated with 100 μM of lansoprazole for 3 h and were then treated with 20 μM of cisplatin. After 24 h, 3-(4,5-dimethylthiazol-2-yl)-5-(3-carboxymethoxyphenyl)-2-(4-sulfophenyl)-2H-tetrazolium, inner salt (MTS) reagent (20 μL/well) (Promega Corp.) was added to each well, and cells were incubated for a further 1 h at 37°C. The absorbance was recorded at 490 nm in a 96-well plate reader (Corona Electric Co., Ltd., Ibaraki, Japan). Data for cell viability represent the results of at least three independent experiments, each of which was performed in triplicate.

### Subcellular fractionation and immunoblots

Nuclear and cytoplasmic fractions were prepared by using a nuclear/cytosol fractionation kit (BioVision Inc., Mountain View, CA, USA). Briefly, cells were harvested with cold phosphate-buffered saline (PBS). Cells were then lysed in 400 μL of the extraction buffer containing dithiothreitol and a protease inhibitor cocktail (Roche, Mannheim, Germany), on ice, for 30 min. After 10 min of centrifugation (700 × g), the supernatant (cytosolic fraction) was

collected, and the pellet was resuspended in a nuclear extraction buffer to obtain the nuclear fraction. Whole cell extracts were prepared with a RIPA buffer (50 mM Tris-HCl pH 7.6, 150 mM of NaCl, 1% NP-40, 0.5% sodium deoxycholate, and 0.1% SDS) (Nacalai Tesque, Inc., Kyoto, Japan) with a protease inhibitor cocktail (Roche). The protein concentration was determined using a BCA protein assay kit (Takara Bio, Inc., Shiga, Japan). Cell lysates (30 μg protein/lane) were resolved using 10% SDS-polyacrylamide gel electrophoresis (SDS-PAGE), and separated proteins were transferred to a polyvinylidene difluoride membrane (Merck Millipore, Bedford, MA, USA). After blocking with 5% nonfat dry milk for 1 h at room temperature, the membrane was incubated overnight at 4˚C with a primary antibody. The following primary antibodies were used: anti-Nrf2 antibody (1:5,000, 16396-1-AP; Proteintech Group, Inc., Rosemont, IL, USA), anti-Histone H3 antibody (1:1,000, ab1791; Abcam, Cambridge UK), anti-α-tubulin antibody (1:1000, sc-23948; Santa Cruz Biotechnology), anti-β-actin antibody (1500, A5060; Sigma-Aldrich), anti-HO1 antibody (13000, SPA-895; Enzo Life Sciences, Inc., Farmingdale, New York, USA), anti-p38α/β (A-12) antibody (1:1,000, sc-7972; Santa Cruz Biotechnology), anti-phospho-p38 antibody (1:1,000, no. 9211; Cell Signaling Technology, Inc., Danvers, MA, USA), anti-ERK1/2 antibody (1:1,000, no. 4695; Cell Signaling Technology, Inc.), anti-phospho-ERK1/2 antibody (1:1,000, no. 9101; Cell Signaling Technology, Inc.), anti-JNK antibody (1:1,000, no. 9252; Cell Signaling Technology, Inc.), and anti-phospho-JNK antibody (1:1,000, no. 9251; Cell Signaling Technology, Inc.). Following incubation with an appropriate secondary antibody for 1 h at room temperature, the bound antibodies were detected using a Western BLoT Hyper HRP Substrate (Takara Bio, Inc.). Three independent experiments were performed, and representative immunoblot analysis of the experiment is presented below.

### Cycloheximide chase assay

Cells were treated with 100 μM of lansoprazole for 3 h followed by incubation with 10 μM of CHX. Cell lysates were prepared 0, 15, 30, 45, 60, and 75 min after the CHX treatment. Whole cell extracts (10 μg) were resolved using SDS-PAGE and were analyzed by immunoblotting with an anti-Nrf2 antibody (Proteintech Group, Inc.); an anti-β-actin antibody (Sigma-Aldrich) was used as a loading control. The degradation rate of the Nrf2 protein was quantified by a densitometric measurement of the immunoblot intensity with ImageJ ver. 1.53v 21 (U.S. National Institutes of Health, Bethesda, Maryland, USA). Data represent the results of three independent experiments, each of which was performed in triplicate.

### λ protein phosphatase (λPPase) assay

For dephosphorylation reactions, the nuclear extracts obtained from lansoprazole-treated cells were prepared as described above. Extracts were incubated with λPPase (New England BioLabs Japan Inc., Tokyo, Japan) at 8 units of enzyme/μg of protein extract diluted in 1×NEBuffer (50 mM of HEPES, 100 mM of NaCl, 2 mM of DTT, and 0.01% Brij 35 at pH 7.5) and 1 mM of $MnCl_2$ at 30˚C, for 30 min. Following incubation, samples were analyzed by immunoblotting. Three independent experiments were performed, and representative immunoblot analysis of the experiment is presented below.

### RNA extraction and quantitative RT-PCR

Total RNA was isolated from cultured cells using Sepasol reagent (Nacalai Tesque, Inc.). For all samples, 200 ng total RNA was used to make cDNA. First strand cDNA was synthesized with the use of the ReverTra Ace qPCR RT Kit (Toyobo, Osaka, Japan) following the manufacturer's instructions in a final volume of 20 μl. The obtained cDNA fragments were diluted

**Table 1. The primer sequences used for quantitative RT-PCR.**

| Gene | Primer | Sequence (5′–3′) |
|---|---|---|
| HO1 (rat) | Forward | ACAGGGTGACAGAAGAGGCTAA |
|  | Reverse | CTGTGAGGGACTCTGGTCTTTG |
| NQO1 (rat) | Forward | CAGCGGCTCCATGTACT |
|  | Reverse | GACCTGGAAGCCACAGAAG |
| GSTA2 (rat) | Forward | CTTCTCCTCTATGTTGAAGAGTTTG |
|  | Reverse | TTTTGCATCCACGGGAA |
| β-actin (rat) | Forward | GGAGATTACTGCCCTGGCTCCTA |
|  | Reverse | GACTCATCGTACTCCTGCTTGCTG |
| GAPDH (rat) | Forward | AGGTTGTCTCCTGTGACTTC |
|  | Reverse | CTGTTGCTGTAGCCATATTC |

10-fold with water to 1 ng/μl. Quantitative RT-PCR analyses were performed using the Brilliant III Ultra-Fast SYBR Green QPCR Master Mix (Agilent Technologies Inc., Tokyo, Japan) and the AriaMX Real-time PCR system (Agilent Technologies Inc.). Each reaction contained 2.5 μL diluted cDNA (2.5 ng/reaction), 200 nM of each primer, and 1 × SYBR Green Master Mix, in a final volume of 10 μL. All reactions were performed in duplicate per cDNA sample. As a control for genomic DNA contamination, total RNA without reverse transcription was tested for each sample per gene. A no-template control was included in each run per gene. The thermal profile of the reaction was 95°C for 5 min activation and denaturation, followed by 40 cycles of 95°C for 10 sec, and 60°C for 10 sec. Finally, a melting curve was generated by increasing the temperature starting from 65°C to 95°C to determine the specificity of the reactions. The quantification cycle number (Cq) was determined per reaction with Agilent AriaMx version 2.0. The primers used are listed in Table 1. The relative standard curve method was used to calculate the relative mRNA expression. mRNA expression levels were normalized to those of β-actin and glyceraldehyde-triphosphate dehydrogenase (GAPDH) mRNA. Data represent the results of three independent experiments, each of which was performed in triplicate.

## siRNA transfection

Two stealth siRNA duplexes targeting the rat Nrf2 mRNA (#1: 5'-UGG AGC AAG ACU UGG GCC ACU UAA A-3'; #2: 5'-GGA AAC CUU ACU CUC CCA GUG AGU A-3') and the stealth RNAi-negative control siRNA (medium GC content) were purchased from Thermo Fisher Scientific, Inc. (Waltham, MA, USA). Cells were transfected with siRNA (5 nM) using Lipofectamine RNAiMax and Opti-MEM media (Thermo Fisher Scientific, Inc., Waltham, MA, USA) according to the manufacturer's instructions. Data represent the results of three independent experiments, each of which was performed in triplicate.

## Generation of a stable ARE-driven reporter system

The ARE-driven reporter gene construct pGL4.37 [luc2P/ARE/Hygro] was purchased from Promega Corp (Madison, WI, USA). RL34 cells were transfected with the pGL4.37 plasmid using jetPRIME transfection reagent. After incubation with growth media for 48 h, cells were cultured with media containing 400 μg/mL of hygromycin B (Sigma-Aldrich, St. Louis, MO, USA) for 3 weeks in order to establish a stable cell line. Hygromycin B-resistant colonies were pooled and used for the reporter gene analysis. Stable transfectants were maintained in DMEM supplemented with 10% FBS and 80 μg/mL hygromycin B at 37°C, with 5% $CO_2$.

## Measurement of ARE-dependent transcriptional activity

Cells stably transfected with an ARE-reporter were replated in 12-well plates and exposed to 100 μM of lansoprazole for 3 h. Luciferase activities in cell lysates were measured using the luciferase assay system (Promega Corp.). Data represent the results of at least three independent experiments, each of which was performed in triplicate.

## Immunocytochemistry

Cells were grown on glass bottom dishes (MatTek Corporation, Ashland, MA, USA) and exposed to 100 μM of lansoprazole for 3 h. Cells were then washed with PBS and fixed in 4% (w/v) paraformaldehyde (PFA) in phosphate buffer, at room temperature. Fixed cells were then permeabilized in 0.1% Triton X-100. After blocking with 1% bovine serum albumin for 30 min at room temperature, cells were incubated with an anti-Nrf2 (H-300) antibody (1:50, sc-13032; Santa Cruz Biotechnology, Inc., Dallas, TX, USA) overnight, at 4˚C. Following three washes in PBS, an Alexa Fluor 488 goat anti-rabbit IgG antibody (1:250; Thermo Fisher Scientific, Inc., Waltham, MA, USA) was added for 2 h, at room temperature. Finally, cells were stained with Hoechst 33342 dye (Dojindo Laboratories, Kumamoto, Japan). Images were obtained using a confocal microscope (Carl Zeiss Japan, LSM 700 ZEN, Tokyo, Japan). Three independent experiments were performed, and representative immunocytochemical analysis of the experiment is shown.

## Statistical analysis

All statistical analyses were performed using JMP version 14.3 statistical software (SAS Institute Inc., Cary, NC, USA). Results are expressed as mean ± standard deviation (SD). Statistical analyses were performed using an unpaired Student's $t$-test. For multiple comparisons, we performed one-way analysis of variance (ANOVA) followed by Tukey's multiple comparison *post hoc* test. Values of $p$ that were found to be <0.05 were considered to be statistically significant.

# Results

## Lansoprazole activates the Nrf2/ARE pathway in RL34 cells

A series of experiments using RL34 cells, a rat hepatic cell line was designed to investigate the molecular mechanism underlying the activation of the Nrf2 pathway. We first examined whether lansoprazole could activate the Nrf2/ARE pathway in this cell line. Immunoblotting studies confirmed the significant increases in Nrf2 levels in both the nuclear (41-fold) and the cytoplasmic (3-fold) fractions of the lansoprazole-treated cells (Fig 1A). Subsequently, to confirm the nuclear localization of the Nrf2 protein following treatment with lansoprazole, immunocytochemistry with an anti-Nrf2 antibody was performed. Nrf2 exhibited weak cytoplasmic staining in the control (DMSO-treated) cells, whereas the lansoprazole-treated cells exhibited strong nuclear Nrf2 staining (Fig 1B). Furthermore, the mRNA levels of the Nrf2 target genes encoding *HO1*, *NQO1*, and *GSTA2* were increased by 17-, 12- and 33-times, respectively, when compared with those of the DMSO-treated cells in the results using *β-actin* as a reference (Fig 1C). Similar results were obtained when using *GAPDH* as a reference gene (Fig 1D). Two clonal RL34 cell lines stably expressing an ARE-driven luciferase reporter (clones #1 and #2) were then established to analyze the transcriptional activity of the ARE promoter. These cells exhibited 20- and 30-fold induction of their luciferase activity following treatment with lansoprazole for 3 h (Fig 1E). Lansoprazole is suggested by these results to induce the Nrf2 nuclear accumulation, and subsequently enhance the expression of antioxidant genes in RL34 cells. Nrf2 activators are recognized to increase the stability of the Nrf2 protein by inhibiting the

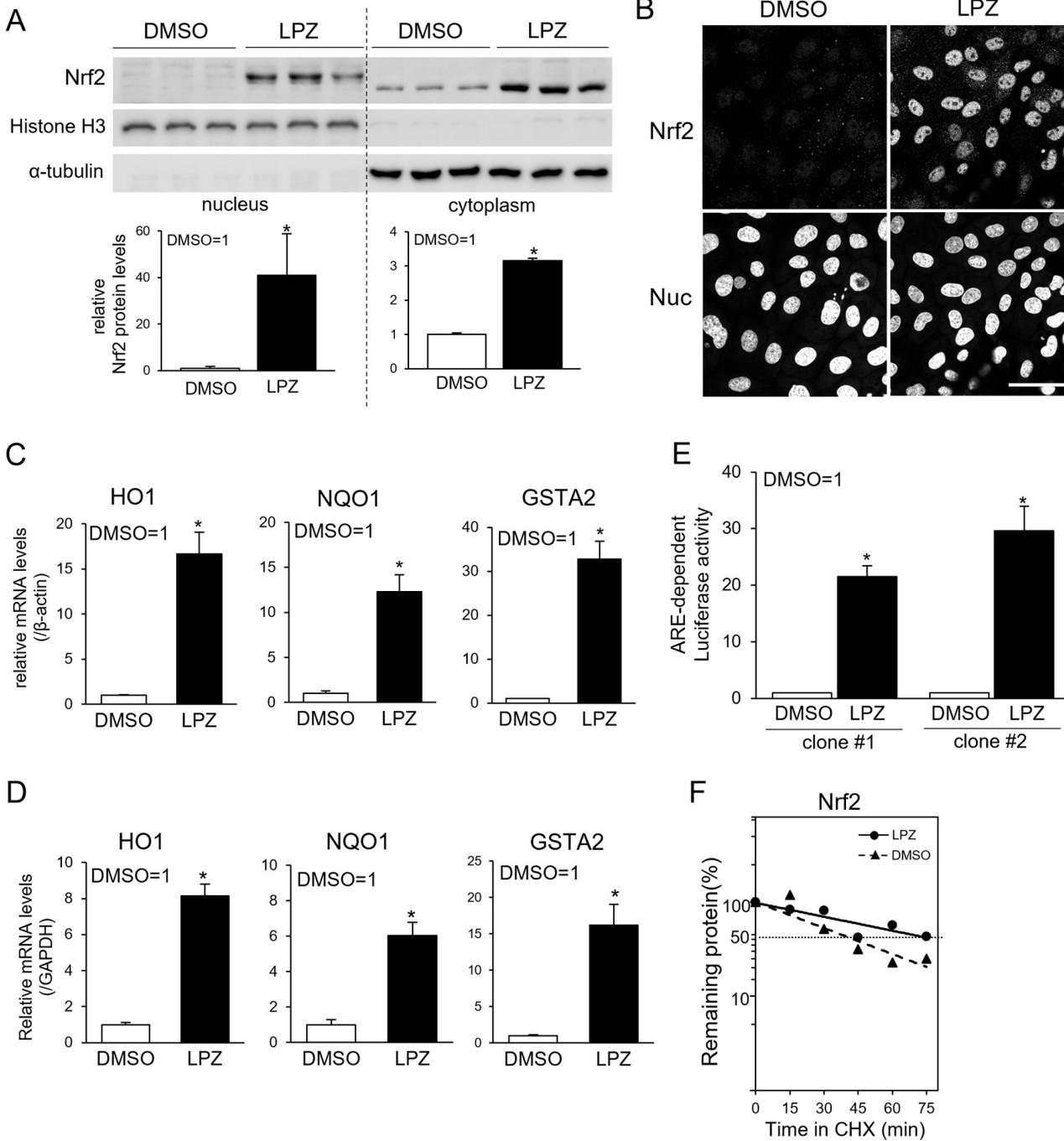

**Fig 1. Lansoprazole activates the Nrf2/ARE pathway in RL34 cells.** A: Representative immunoblot for Nrf2 expression levels in the nuclear and cytoplasmic protein lysates obtained from cells treated with 100 μM of lansoprazole for 3 h; α-tubulin and histone H1 were loading controls for the cytoplasmic and nuclear lysates, respectively (left). Densitometric quantification of the representative immunoblots was performed using ImageJ, and values were normalized to each of the loading control protein level (right). B: Representative immunocytochemical images of the subcellular localization of Nrf2. RL34 cells were treated with 100 μM of lansoprazole for 3 h. Cells were fixed in 4% PFA for 15 min, followed by staining with an anti-Nrf2 antibody and an Alexa Fluor 488-conjugated goat anti-rabbit IgG antibody. Counterstaining of the nuclei was performed with Hoechst 33342. The scale bar represents 50 μm. C and D: Relative mRNA expression levels of Nrf2-induced genes *HO1*, *NQO1*, and *GSTA2* in cells treated with 100 μM lansoprazole for 3 h. RNA levels were normalized to those of *β-actin* (C) or *GAPDH* (D). E: Induction of the ARE-dependent luciferase reporter activity by lansoprazole. RL34 cells stably expressing an ARE-reporter gene were treated with 100 μM of lansoprazole for 3 h, and the luciferase activity in the cell lysates was measured. F: Degradation rate of the Nrf2 protein after CHX chase. Cells were treated with 100 μM of lansoprazole for 3 h and were then treated with 10 μM of CHX for the indicated periods. Cell lysates were used for immunoblotting analysis with an anti-Nrf2 and anti-β-actin antibody. Intensities of the Nrf2 bands were quantified and plotted on a semilog graph to obtain half-life values. Abbreviations used in the figure: LPZ, lansoprazole; CHX, cycloheximide. Data represent the mean ± SD of the three independent experiments performed in triplicate. Statistical analysis was performed by using Student's *t*-test. *: $p < 0.05$ *vs*. DMSO-treated cells.

proteasomal degradation of Keap1 [34, 35]. The effect of lansoprazole on the degradation rate of the Nrf2 protein was therefore determined using a CHX chase assay. Treatment with lansoprazole extended the half-life of the Nrf2 protein from 33.0 to 61.5 min (Fig 1F). Lansoprazole is indicated to inhibit the degradation of the Nrf2 protein, and increases its stability.

## Lansoprazole suppresses the cisplatin-induced cytotoxicity; activity is partially dependent upon HO1

An *in vitro* cisplatin-induced cytotoxicity model was conducted to assess the effect of lansoprazole on oxidative stress-induced cell death. Cisplatin cytotoxicity was determined using an MTS assay. Cell viability was decreased to 25% by cisplatin treatment when compared with that of untreated cells, whereas cells pretreated with lansoprazole exhibited an increase in their cell viability of approximately 80% (Fig 2). The effect of lansoprazole was partially eliminated by co-treatment with the specific HO1 inhibitor tin-mesoporphyrin IX (SnMP). Lansoprazole is suggested by these results to have a protective effect against the cisplatin-induced cytotoxicity and this effect is partially mediated by the upregulation of HO1.

## Nrf2 is necessary for the cytoprotective activity of lansoprazole

To demonstrate whether the induction of expression of HO1 by lansoprazole is dependent upon that of Nrf2, an siRNA-mediated silencing of Nrf2 was performed. RL34 cells were transfected with one of two siRNAs targeting different sites in the rat *Nrf2* mRNA (siNrf2#1 and siNrf2#2). After 24 h, the *Nrf2* mRNA levels were examined by quantitative RT-PCR. The knockdown efficiencies of the individual siRNAs on the Nrf2 expression were 91% (siNrf2#1) and 91.4% (siNrf2#2) as compared with the expression levels of cells treated with the control siRNA in the results using *β-actin* as a reference (Fig 3A). Similar results were obtained when

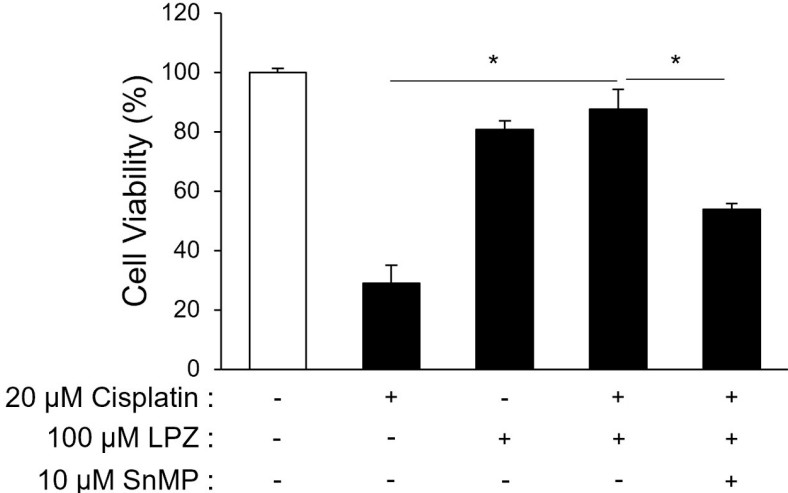

**Fig 2. Lansoprazole suppresses the cisplatin-induced cytotoxicity.** Cell viability of cisplatin-treated cells exposed to a lansoprazole pretreatment. RL34 cells were pretreated with 100 μM of lansoprazole for 3 h and then exposed to 20 μM of cisplatin for an additional 24 h. HO1 inhibitor SnMP was administered simultaneously with lansoprazole. Cell viability was quantified using an MTS assay. Data are expressed as a percentage of viability when compared with the viability of the DMSO-treated cells. Data represent the mean ± SD of three independent experiments performed in triplicate. Statistical analysis was performed by using ANOVA followed by Tukey's test for multiple comparisons. *: $p < 0.01$. Abbreviations: LPZ, lansoprazole; SnMP, tin-mesoporphyrin IX; MTS, 3-(4,5-dimethylthiazol-2-yl)-5-(3-carboxymethoxyphenyl)-2-(4-sulfophenyl)-2H-tetrazolium,inner salt.

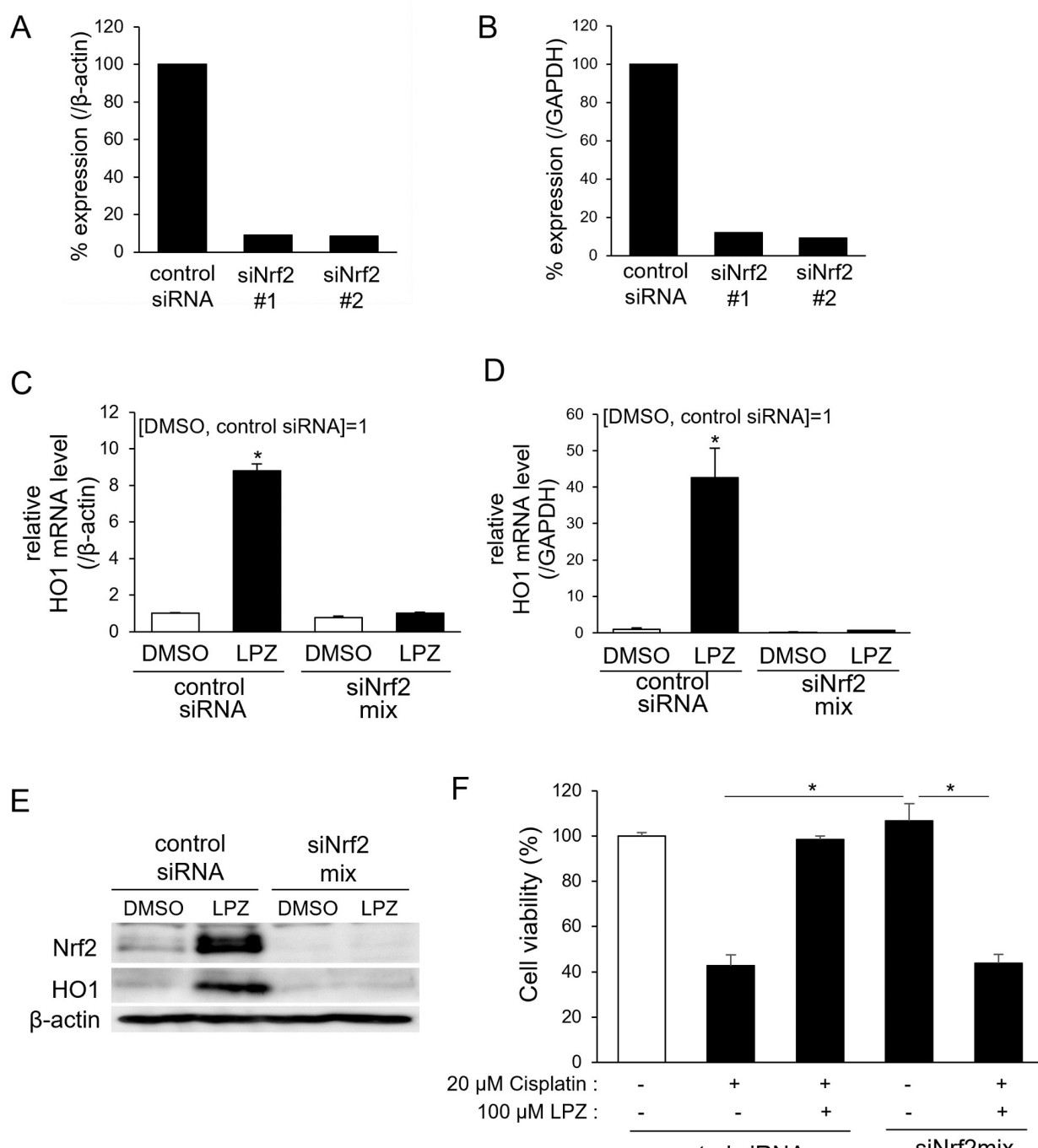

**Fig 3. Nrf2 is essential for the cytoprotective activity of lansoprazole.** A and B: Knockdown efficiency of siRNAs targeting Nrf2. RL34 cells were transfected with one of the two siRNAs targeting different sites in the rat *Nrf2* mRNA (5 nM) or with control siRNA (5 nM) for 24 h. *Nrf2* mRNA levels were measured using quantitative RT-PCR and were normalized to those of *β-actin* (A) or *GAPDH* (B). Results are expressed as a percentage of the mRNA levels of the samples treated with control siRNA. C and D: Effects of Nrf2 knockdown on the expression of the lansoprazole-induced gene *HO1* mRNA. Cells were transfected with 5 nM of control or pooled Nrf2 siRNA and, 24 h later, were treated with 100 μM of lansoprazole for 3 h. *HO1* mRNA levels were measured using quantitative RT-PCR; RNA levels were normalized to the *β-actin* (C) or *GAPDH* (D) mRNA level. Data represent the mean ± SD of three independent experiments performed in triplicate. Statistical analysis was performed using Student's *t*-test. *: $p < 0.05$ *vs*. DMSO-treated control siRNA-transfected cells. E: Effects of Nrf2 knockdown on the expression of lansoprazole-induced Nrf2 and HO1 protein. Cells were transfected with 5 nM of control or pooled Nrf2 siRNA and then 24 h later were treated with 100 μM of lansoprazole for 3 h. Nrf2 and HO1 protein levels were measured using immunoblotting. β-actin served as a loading control. F: Effect of siRNA depletion of Nrf2 on the lansoprazole-mediated cytoprotection. siRNA-transfected cells were treated with 100 μM of lansoprazole

for 3 h and then exposed to 20 μM of cisplatin for an additional 24 h. Cell viability was quantified using an MTS assay. Data are expressed as a percentage of viability when compared with DMSO-treated control siRNA-transfected cells. Data represent the mean ± SD of three independent experiments performed in triplicate. Statistical analysis was performed by using ANOVA and Dunnett's test. *: $p < 0.01$ *vs.* cisplatin treatment group. Abbreviations: LPZ, lansoprazole; MTS, 3-(4,5-dimethylthiazol-2-yl)-5-(3-carboxymethoxyphenyl)-2-(4-sulfophenyl)-2H-tetrazolium, inner salt.

using GAPDH as a reference gene (Fig 3B). RL34 cells were transfected with pooled Nrf2 siR-NAs, and the expression level of lansoprazole-induced HO1 were assessed. As expected, the Nrf2 knockdown completely blocked the lansoprazole-induced expression of both *HO1* mRNA (Fig 3C and 3D) and protein (Fig 3E). Lansoprazole is suggested by these results to induce the expression of *HO1* mRNA and protein in an Nrf2-dependent manner. Further-more, the effect of Nrf2 depletion on lansoprazole-mediated cytoprotection against cisplatin-induced cytotoxicity was assessed. RL34 cells were transfected with siNrf2 pools, followed by treatment with lansoprazole for 3 h. The cisplatin-induced cytotoxicity was then measured using an MTS assay. The cytotoxicity of cisplatin was increased in the Nrf2-knockdown cells to the same level as that observed in cells receiving a single administration of cisplatin (Fig 3F). The cytoprotective effect of lansoprazole is suggested to be dependent upon the Nrf2 pathway.

## Lansoprazole activates the p38 MAPK signaling but not ERK1/2 or JNK signaling

The Nrf2 band was found to be upshifted in the nuclear fraction of the lansoprazole-treated cells (Fig 1A), indicating that Nrf2 translocated to the nucleus underwent posttranscriptional modifications. Phosphorylation is known to induce an upward shift of protein bands, so we examined the phosphorylation state of the Nrf2 in the nucleus. Nuclear fractions of the lanso-prazole-treated cells were incubated in the presence or absence of λPPase, at 4˚C or 30˚C (Fig 4A). The nuclear Nrf2 incubated with phosphatase at 30˚C exhibited a downshift on SDS-PAGE, whereas those incubated without the phosphatase (at 4˚C and at 30˚C) did not, indicating that Nrf2 is phosphorylated in the nucleus. The effect of lansoprazole on the phos-phorylation of three major MAPKs (namely, p38 MAPK, ERK1/2, and JNK) were examined using RL34 cells (Fig 4B). The phosphorylation of p38 MAPK was enhanced in the lansopra-zole-treated cells, whereas that of ERK1/2 and JNK had no apparent changes. Lansoprazole therefore specifically activates p38 MAPK, which subsequently plays an important role in the nuclear translocation of Nrf2 in RL34 cells.

## Activation of p38 MAPK is required for the lansoprazole-induced Nrf2/ ARE pathway in RL34 cells

A selective inhibitor of p38 MAPK (SB203580) was then used to investigate the role of p38 MAPK in the activation of the Nrf2/ARE pathway in RL34 cells [36]. First, the effect of p38 MAPK on the Nrf2 upregulation by lansoprazole was examined. Immunoblotting analysis revealed that the lansoprazole-induced Nrf2 upregulation was completely inhibited by the addition of SB203580 (Fig 5A). Then inhibition of p38 MAPK by SB203580 completely can-celed the lansoprazole-induced ARE-associated luciferase activity (Fig 5B) and the expression of HO1 (Fig 5C and 5D). Consistent with these data, immunofluorescence analysis revealed that the lansoprazole-induced nuclear translocation of Nrf2 was impaired by SB203580 (Fig 5E). SB203580 was suggested by these findings to have blocked the Nrf2/ARE pathway upstream of the nuclear translocation of Nrf2. p38 MAPK is therefore likely the key MAPK whose activation is required for the lansoprazole-induced HO1 expression in RL34 cells.

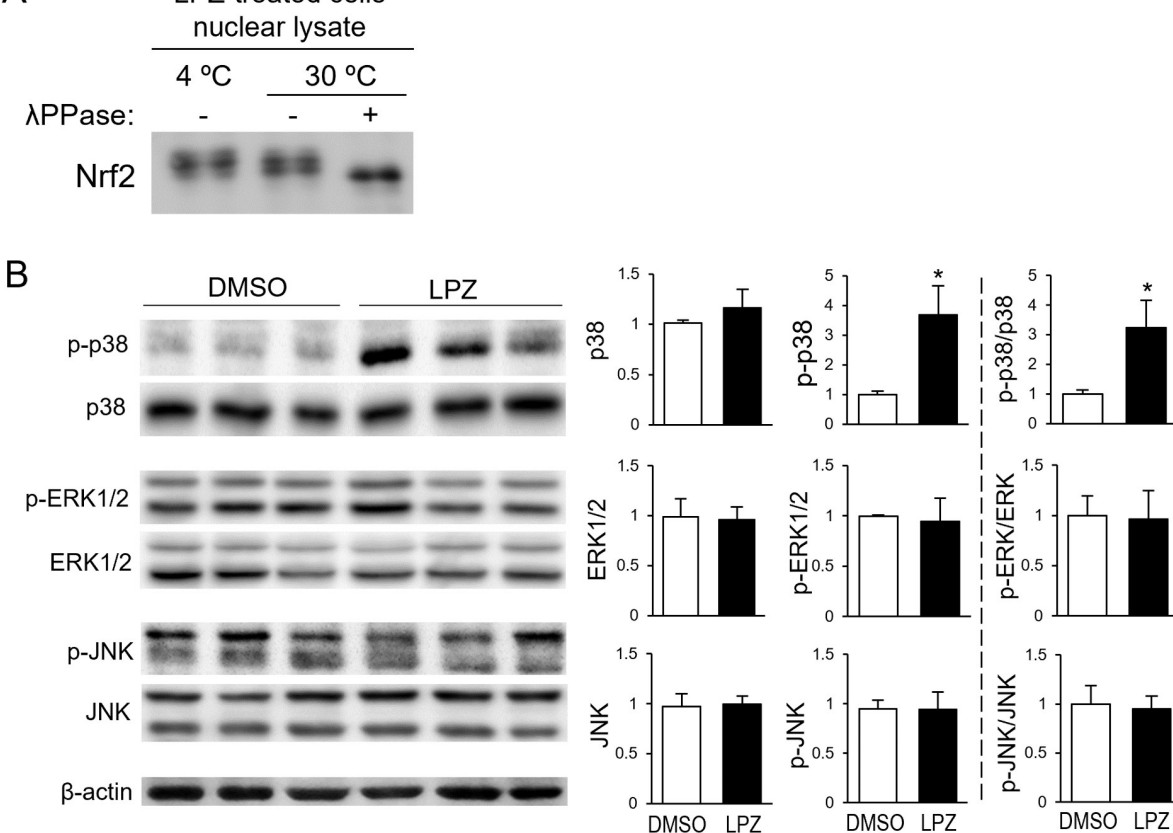

**Fig 4. Lansoprazole activates p38 MAPK, but not ERK1/2 or JNK in RL34 cells.** A: Detection of phosphorylated Nrf2 in the nucleus. RL34 cells were treated with lansoprazole for 3 h. Nuclear protein extracts from the cells were incubated with or without λPPase at 4°C or 30°C, for 30 min, and were analyzed using immunoblotting with an anti-Nrf2 antibody. B: Representative immunoblot for the phosphorylated and total p38 MAPK, ERK1/2, and JNK expression levels. RL34 cells were treated with 100 μM of lansoprazole for 3 h. Whole cell lysates from the cells were analyzed using immunoblotting with the indicated antibodies; β-actin served as a loading control (left). Densitometric quantification of representative immunoblots was performed using ImageJ. Values were normalized to β-actin protein levels. The ratio of the phosphorylated protein to total protein were also calculated (right). Data represent the mean ± SD of three independent experiments performed in triplicate. Statistical analysis was performed using Student's *t*-test. *: $p < 0.05$ *vs*. DMSO-treated cells. Abbreviation: LPZ, lansoprazole.

### Activation of p38 MAPK is required for the cytoprotective activity of lansoprazole against cisplatin-induced toxicity

We subsequently investigated whether the p38 MAPK pathway is involved in the cytoprotective activity of lansoprazole. RL34 cells were treated with lansoprazole for 3 h in the presence or absence of an SB203580 pretreatment and the MTS assay was used to determine the cell viability after exposure to cisplatin for 24 h. The inhibition of the p38 MAPK pathway by SB203580 completely restored the sensitivity to cisplatin in RL34 cells treated with lansoprazole (Fig 6). Cytoprotective activity of lansoprazole is therefore suggested to be through the p38 MAPK pathway.

## Discussion

The present study demonstrated that lansoprazole can activate the Nrf2/ARE-mediated antioxidant and phase 2 detoxifying gene expression, and can suppress cisplatin-induced

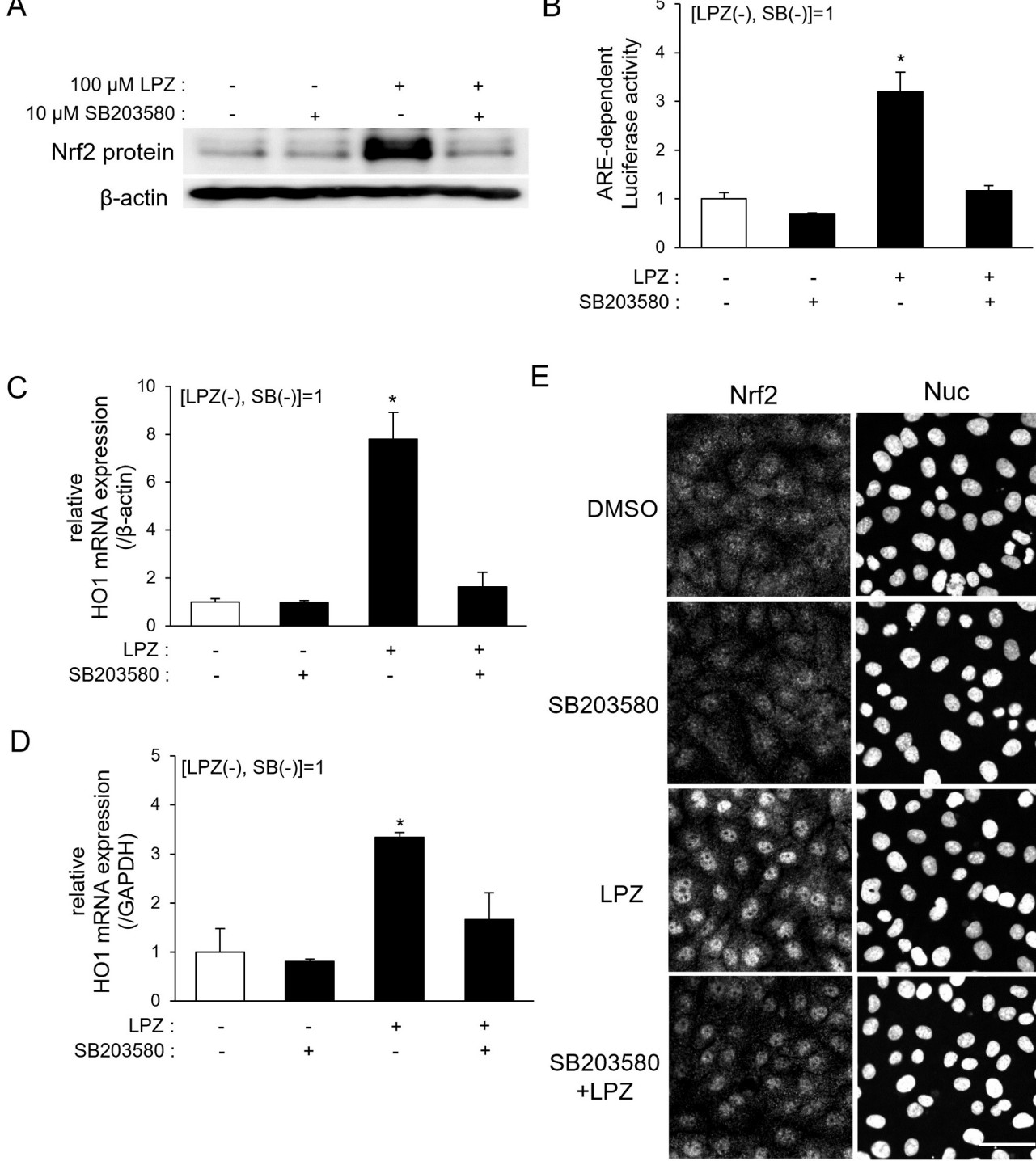

**Fig 5. Activation of p38 MAPK is required for the lansoprazole-induced ARE/Nrf2 pathway in RL34 cells.** A: Effect of SB203580 on the induction of Nrf2 protein expression by lansoprazole. RL34 cells were treated with 10 μM of SB203580 (a specific p38 MAPK inhibitor) for 30 min, and then exposed to 100 μM of lansoprazole for an additional 3 h. Whole cell protein lysates were analyzed by immunoblotting with an anti-Nrf2 antibody; β-actin served as a loading control. B: Effect of SB203580 on the induction of the ARE-dependent reporter activity by lansoprazole. RL34 cells stably expressing the ARE-reporter gene were pretreated with 10 μM of SB203580 for 30 min, and then exposed to 100 μM of lansoprazole for an additional 3 h. Luciferase activity in the cell lysates was measured. C and D: Total RNA was isolated from the cells. Relative mRNA expression levels of *HO1* were measured by quantitative RT-PCR. RNA levels were normalized to *β-actin* (C) and *GAPDH* (D) mRNA level. Data represent the mean ± SD of three independent experiments performed in triplicate. Student's *t*-test was employed for statistical analysis. *: $p < 0.05$ *vs.* untreated cells. E: Representative immunocytochemical images of the subcellular localization of Nrf2. RL34 cells were treated with 10 μM of SB203580 for 30 min and then exposed to

100 µM of lansoprazole for an additional 3 h. Cells were fixed in 4% PFA for 15 min, followed by a staining with anti-Nrf2 antibody and an Alexa Fluor 488-conjugated goat anti-rabbit IgG antibody; counterstaining of the nuclei was performed with Hoechst 33342. The scale bar represents 50 µm. Abbreviation: LPZ, lansoprazole.

cytotoxicity. These effects were completely canceled by knockdown of Nrf2 and inhibition of p38 MAPK, suggesting that the function of lansoprazole for inhibiting cisplatin-induced cell death is dependent on Nrf2 and p38 MAPK. In addition, the Nrf2 protein accumulated in the nucleus following lansoprazole treatment was phosphorylated, suggesting that this modification is involved in the anti-cell death function of lansoprazole. We previously reported that lansoprazole upregulates Nrf2 expression and attenuates the drug-induced oxidative liver injury in rats [32, 33]. Lansoprazole has been reported to exhibit Nrf2-mediated antioxidant and anti-inflammatory effects in gastric mucosal cells [23] and the small intestine [28] and in the rat kidneys [37]. Our data demonstrated that lansoprazole is a potent inducer of the Nrf2/ARE antioxidant pathway in hepatic cells, independently of its acid secretory-related actions (Fig 7). Lansoprazole is one of the most widely prescribed drugs worldwide, and is considered to have well-established safety and efficacy. Clinical application of lansoprazole is therefore expected to expand relatively quickly in the future as a treatment for liver diseases.

Cisplatin is a commonly utilized chemotherapeutic agent, and confers toxicity to various tissues, including the liver. Growing evidence indicates that the production of ROS induced by cisplatin can trigger oxidative stress, and even cause cell death [16–18, 38, 39]. Increased oxidative stress is one of the main mechanisms involved in cisplatin-induced hepatotoxicity, as indicated by both *in vitro* and *in vivo* studies [18]. The activation of the Nrf2 signaling pathway has been associated with an increase in the expression of antioxidant enzymes and a decrease in ROS levels [40–42]. An Nrf2 activator should therefore be expected to provide organ and cell protection against cisplatin-induced cytotoxicity. Consistent with previous reports, this study demonstrated that lansoprazole confers protective effects against cisplatin-induced cytotoxicity in RL34 cells. Marullo *et al.* showed that cisplatin induced direct damage to mitochondrial DNA, causing oxidative stress by increasing the intracellular ROS level [17]. Mitochondria are

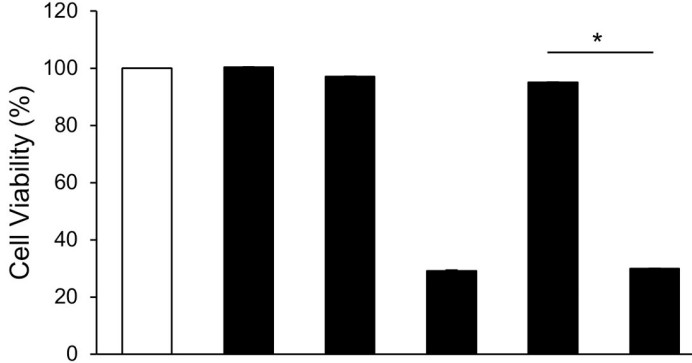

**Fig 6. Activation of p38 MAPK is required for the cytoprotective activity of lansoprazole against cisplatin-induced toxicity.** Effect of SB203580 on the cytoprotective activity of lansoprazole. RL34 cells were treated with 100 µM of lansoprazole for 3 h in the presence or absence of 10 µM of an SB203580 (a specific p38 MAPK inhibitor) pretreatment for 30 min and were then exposed to 20 µM of cisplatin for an additional 24 h. Cell viability was quantified using an MTS assay. Data are expressed as a percentage of viability of nontreated cells. Data represent the mean ± SD of three independent experiments performed in triplicate. Statistical analysis was performed using ANOVA followed by Tukey's test for multiple comparisons. *: $p < 0.01$. Abbreviations: LPZ, lansoprazole; MTS, 3-(4,5-dimethylthiazol-2-yl)-5-(3-carboxymethoxyphenyl)-2-(4-sulfophenyl)-2H-tetrazolium, inner salt.

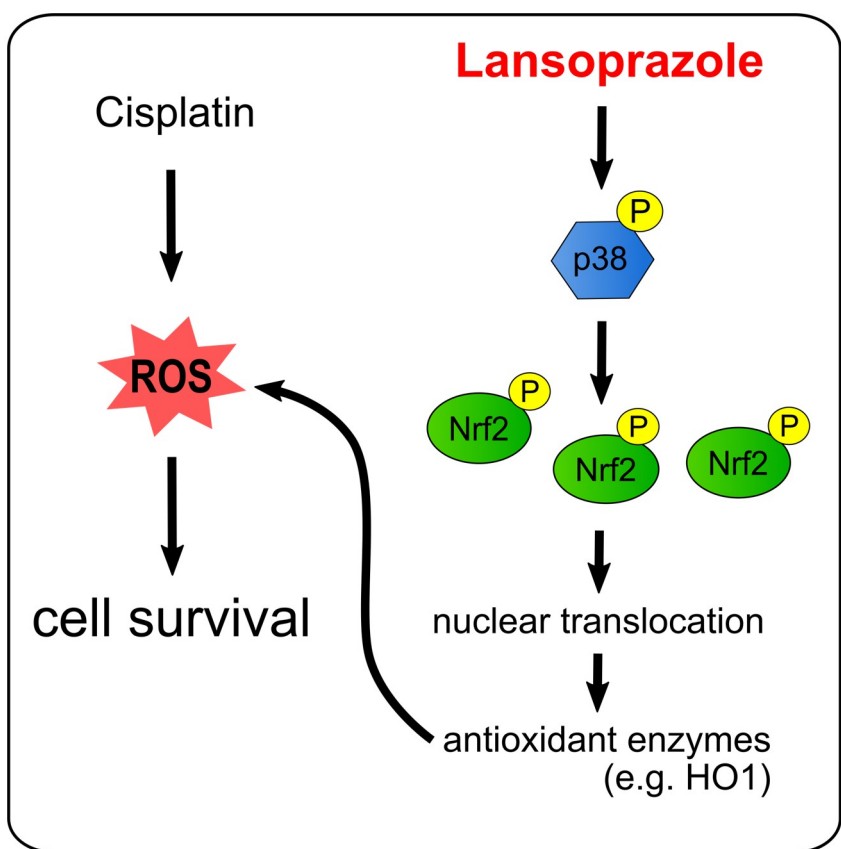

**Fig 7. Lansoprazole activates the p38 MAPK/Nrf2/ARE pathway and suppresses cisplatin-induced cell death in RL34 cells.** Schematic model of the Nrf2-dependent cytoprotective signaling pathway by lansoprazole in RL34 cells. Lansoprazole suppresses cisplatin-induced cytotoxicity through the activation of the Nrf2 pathway, accompanied by induction of antioxidant genes such as that encoding HO1. These effects are dependent on the p38 MAPK pathway.

the main source of ROS, so it can be inferred that lansoprazole can scavenge exogenous or endogenous ROS other than that induced by cisplatin. Lansoprazole could have wider clinical applications. The cisplatin-induced cytotoxicity model using RL34 cells employed in this study has been a useful model for the establishment of the cytoprotective mechanism of lansoprazole.

Nrf2 plays a crucial role in regulating cellular redox homeostasis, and protects cells from oxidative stress by inducing the transcriptional activation of antioxidant and phase II detoxifying enzymes through the ARE [43–45]. Nrf2 protection of the liver from several hepatotoxicants has been shown in many studies, and this is known to be accompanied by induction of antioxidant genes encoding proteins, such as HO1 [3]. Nrf2 activation was predicted to play a crucial role in the cytoprotective effect of lansoprazole. As expected, the siRNA-mediated knockdown of Nrf2 completely abolished the upregulation of HO1 as well as the cytoprotective effect of lansoprazole, whereas SnMP, an inhibitor of HO1, was able to partially block this effect. These results are inconsistent with those of previous studies [28], and the induction of the HO1 expression did not fully account for the protective effect of lansoprazole. Lansoprazole has been observed to significantly increase the expression levels of *NQO1* and *GSTA2* mRNA. Furthermore, these Nrf2-regulated antioxidant genes may contribute to the cytoprotection of lansoprazole. One limitation of this study, based on previous reports [32], is that

sampling was conducted at 3 h after lansoprazole treatment, and stronger antioxidant activity could be observed if tested over a longer time course.

In this study, we attempted to elucidate the intracellular signal transduction mechanism of the hepatic protection by lansoprazole, and we subsequently used a rat hepatic RL34 cell culture model. Using *in vitro* phosphatase assay with RL34 cells, we showed that Nrf2 translocated to the nucleus (as a result of the lansoprazole treatment) is phosphorylated. Phosphorylation of Nrf2 is indicated to be involved in Nrf2 nuclear translocation and the activation of the ARE-associated gene transcription [46, 47]. MAPK signaling (through p38 MAPK, ERK1/2, and JNK) may play an important role in the stress response, including the regulation of the Nrf2/ARE pathway [48–52]. However, the involvement of MAPKs in the lansoprazole-induced pathway remains unclear. Takagi *et al.* reported the role of ERK in lansoprazole-induced expression of HO1 in rat gastric mucosal cell lines [23], whereas Schulz-Geske *et al.* reported that lansoprazole-induced expression of HO1 without an activation of the MAPK pathways in NIH3T3 mouse embryonic fibroblasts [25]. Remarkably, in RL34 cells, lansoprazole could induce the phosphorylation of p38 MAPK, but not that of ERK1/2 or JNK. Further experiments using SB203580, a specific inhibitor for p38 MAPK, confirmed the exclusive role of p38 MAPK in the lansoprazole-induced Nrf2/HO1 activation and cytoprotective effects. However, as a second limitation, our study could not demonstrate whether the p38 MAPK directly phosphorylates the Nrf2 in lansoprazole-treated cells. Further studies are required to define the direct association between p38 MAPK and Nrf2. As a crosstalk between different MAPK signaling pathways does not seem to occur, the RL34 cell line is a useful *in vitro* model for the study of the intracellular molecular mechanism of the lansoprazole-mediated Nrf2 pathway induction in hepatocytes. A third limitation is that the experiments were performed in only one cell line. Although lansoprazole caused exclusive p38 MAPK activation in RL34 cells, it is possible that other MAPK signaling pathways are activated in other cell types (e.g., human). Lansoprazole has in fact been reported to induce HO1 in gastric mucosal cell lines via the ERK pathway [23] and in NIH3T3 mouse embryonic fibroblasts via the PI3K (the phosphatidylinositol 3-kinase) pathway, not the MAPK family [25].

In basal conditions, Nrf2 activity is suppressed by the cytoplasmic repressor Keap1, which promotes its proteasome-dependent degradation. Most Nrf2 inducers are known to react with the cysteine residues of Keap1, causing a conformational change of the Keap1-Nrf2 complex [6] and, thereby, obstructing the Nrf2 degradation. The newly generated Nrf2 accumulates and translocates into the nucleus. Takagi *et al.* showed that stimulation with lansoprazole can induce the oxidation of the Keap1 cysteine residues in gastric mucosal cells [23]. In the current study, we found that treatment with lansoprazole can stabilize the Nrf2 protein. Lansoprazole was suggested to activate the Nrf2/ARE pathway by modifying the interaction between Keap1 and Nrf2, similarly to other Nrf2 inducers. However, as a fourth limitation, this study did not detect a direct modification of the Keap1 protein by lansoprazole. Further studies are required to demonstrate whether the hepatoprotective effect of lansoprazole results from a direct inactivation of Keap1 or from the regulation of the Keap1 inactivation by other upstream factors.

Lansoprazole is a prodrug that needs activation in an acidic environment such as the acidic secretory canaliculi for its anti-secretory effect [53]. Our previous study showed that proton pump H+/K+-adenosine triphosphatase was not expressed in rat liver, and the cytoprotective effect of lansoprazole is therefore assumed to be mediated by an inactive form [32]. Moreover, lansoprazole is metabolized mainly by CYP2C19 and CYP3A4 in the liver [54], with the major metabolites being 5'-hydroxy lansoprazole and the lansoprazole sulfone. As a fifth limitation, the present study could not determine whether the cytoprotective effect of lansoprazole was caused by acid-induced activation or metabolism by CYP enzymes, and this requires future investigation.

The concentration of 100 μM lansoprazole employed in this study is much higher than the maximum serum concentration in healthy adults given a single oral administration of 30 mg lansoprazole, which can be considered as a sixth limitation. However, the plasma concentration of lansoprazole does not correlate with tissue concentrations of lansoprazole. Lansoprazole accumulates in acidic tissue environments, where local concentrations have been proposed to reach millimolar levels [54]. The regular clinical dosage of lansoprazole has been determined by the amount of acid suppression in the stomach. Data regarding optimal dosing is unavailable for *in vivo* studies of Nrf2-mediated hepatoprotective activity. Antioxidant effects on the liver in humans treated with lansoprazole have not been studied, and the optimal dosage requires investigation. As a seventh limitation, lansoprazole is reported to have anti-inflammatory effects [19, 20], but the expression levels of inflammatory cytokines were not investigated in this study. Subsequent studies should therefore focus on inflammatory cytokines and investigate gene expression levels, focusing on those downstream of Nrf2. Furthermore, several reports have shown that lansoprazole can directly scavenge ROS as a substantial scavenger using chemical antioxidant assay [55, 56]. Lansoprazole is suggested to have a direct antioxidant function in the liver, separate from its action via the p38 MAPK/Nrf2 signaling pathway.

In conclusion, our results demonstrate that lansoprazole confers cytoprotection against cisplatin-induced cell death through the activation of the Nrf2/ARE pathway and through the upregulation of antioxidant enzymes downstream of Nrf2 (that takes place *via* the p38 MAPK signaling pathway). Oxidative stress is the cause of most liver diseases, and it is a plausible therapeutic target to prevent liver damage [57]. The induction of the antioxidant Nrf2 pathway may protect hepatic cells from oxidative stress and avoid the progression of disease. Various compounds have been postulated to increase Nrf2 activity, although an effective Nrf2 inducer is not currently available to counteract liver diseases. This study extended prior knowledge on the acid-independent effect of lansoprazole and provided novel insights into the mechanisms involved. Lansoprazole may therefore be a potential candidate for the treatment of liver diseases.

## Supporting information

**S1 Raw images. Original uncropped images underlying all blots.** This file contains all uncropped blot information. The asterisk indicates a nonspecific protein band.
(PDF)

**S1 Data.**
(XLSX)

## Acknowledgments

We acknowledge proofreading and editing by Benjamin Phillis at the Clinical Study Support Center at Wakayama Medical University.

## Author Contributions

**Conceptualization:** Naoko Yamagishi, Yuta Yamamoto.

**Data curation:** Naoko Yamagishi, Toshio Nishi, Takao Ito, Yoshimitsu Kanai.

**Formal analysis:** Naoko Yamagishi.

**Funding acquisition:** Naoko Yamagishi.

**Investigation:** Naoko Yamagishi, Yoshimitsu Kanai.

**Methodology:** Naoko Yamagishi, Yuta Yamamoto, Toshio Nishi.

**Project administration:** Naoko Yamagishi.

**Resources:** Naoko Yamagishi.

**Software:** Naoko Yamagishi.

**Supervision:** Naoko Yamagishi, Yoshimitsu Kanai.

**Validation:** Naoko Yamagishi.

**Visualization:** Naoko Yamagishi.

**Writing – original draft:** Naoko Yamagishi.

**Writing – review & editing:** Naoko Yamagishi, Yuta Yamamoto, Toshio Nishi, Takao Ito, Yoshimitsu Kanai.

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
