## [Decision Letter · Decision Letter 0]

6 Mar 2023

PONE-D-22-35612Activation of p38 MAPK signaling is required for lansoprazole-mediated Nrf2 activation and cytoprotective effects against cisplatin toxicity in rat hepatic RL34 cells.PLOS ONE

Dear Dr. Yamagishi,

Thank you for submitting your manuscript to PLOS ONE. After careful consideration, we feel that it has merit but does not fully meet PLOS ONE’s publication criteria as it currently stands. Therefore, we invite you to submit a revised version of the manuscript that addresses the points raised during the review process.

We look forward to receiving your revised manuscript.

Kind regards,

Wei Hsum Yap

Academic Editor

PLOS ONE

Journal Requirements:

“This work was supported in part by a Japan Society for the Promotion of Science 478 (JSPS) KAKENHI grant (grant no. 17K15963) to N. Yamagishi.”

“This work was supported in part by a Japan Society for the Promotion of Science (JSPS) KAKENHI grant (grant no. 17K15963) to Naoko Yamagishi. The funders had no role in study design, data collection and analysis, decision to publish, or preparation of the manuscript.”

6. Please amend either the title on the online submission form (via Edit Submission) or the title in the manuscript so that they are identical.

Additional Editor Comments:

Please revise the manuscript to enhance clarity of the submission.

Reviewers' comments:

Reviewer's Responses to Questions

**Comments to the Author**

1. Is the manuscript technically sound, and do the data support the conclusions?

Reviewer #1: Yes

Reviewer #2: Partly

2. Has the statistical analysis been performed appropriately and rigorously? 

Reviewer #1: I Don't Know

Reviewer #2: Yes

3. Have the authors made all data underlying the findings in their manuscript fully available?

Reviewer #1: No

Reviewer #2: Yes

4. Is the manuscript presented in an intelligible fashion and written in standard English?

Reviewer #1: Yes

Reviewer #2: Yes

5. Review Comments to the Author

Reviewer #1: This manuscript is very interesting and poses lansoprazole as a potential therapeutic target for oxidative stress to the liver. However, the data could be presented in a more clear fashion, and the wording in some areas could be improved.

Reviewer #2: Overview:

Generally, the manuscript presented the novelty of the study in elucidating the potential molecular mechanisms of lansoprazole in cisplatin-induced hepatoxicity in rat hepatic RL34 cells. This manuscript provides an insightful scientific impact on the molecular mechanisms of lansoprazole. The study design and method used were reliable, yet the reporting in methods and results require improvements.

Some recommendations have been suggested as below for revision.

Strengths:

1. This paper presented novel insights on the mechanisms of lansoprazole.

2. The method used was robust and reliable.

Limitations:

1. Poor reporting in methodology and result.

2. Gene expression study with 1 housekeeping gene is less convincing.

Title: Title is too length with too many unnecessary words. Suggest to revise.

Abstract:

1. Line 14; 28: Kindly avoid the use of first- and second-person pronouns (i.e. we) in the abstract. Please use active voice and third-person narrative throughout the manuscript.

2. Line 20: Typo for ‘NAD(P)H:1quinone oxidoreductase-1’? It should be written as ‘NAD(P)H quinone oxidoreductase-1’.

3. Method was not mentioned in the abstract.

Introduction

1. Line 48: ‘The nuclear Nrf2…’

2. Line 72 - 79: Kindly avoid the use of first- and second-person pronouns (i.e. we) in the abstract. Please use active voice and third-person narrative.

3. Suggest to include the relationship of nrf-2 with p38 and Erk1 in introduction to provide an better overview on the signaling pathway involved and further justified on the choice of antibodies used in Western blotting.

Methods

1. Some chemicals or materials only provided with brand name without the city and country of manufacturing, including FBS, DMSO, DMEM, MTS reagent, lansoprazole, RIPA buffer, hygromycin, luciferase assay, anti-rabbit IgG antibody.

2. How many replicates run in cell viability assay (MTS)? How many sets of experiment samples per replicate? Please mention in the method.

3. Line 106: What 96-well plate reader was used? Please specify the brand of equipment along with the city and country.

4. Line 134: CHX abbreviation was used without providing the full term when it was first introduced in the text.

5. Line 140: Version, city and country of ImageJ?

6. Line 151: The amount of total RNA used to reverse-transcribed into cDNA was not mentioned.

7. The experiment setting for gene expression in PCR assay was unclear: volume of reaction mix of PCR, thermal cycling condition, number of replicates, negative control, the validation or optimization of primers, etc.

8. Line 157: Only one housekeeping gene was used (beta actin), which is insufficient and at least two housekeeping genes are require for in vitro experiments. Based on MIQE guideline for qPCR, normalization of samples obtained from in vitro experiments can be carried out against a panel (probably two or three) of housekeeping genes whose expression has been shown to be unaffected by experimental conditions. This information must be included in any publication. Refer to MIQE guideline 5th edition published in January 2022. http://www.econferences.de/miqe-talks/

9. Line 194: What software was used to run the statistical analyses? Please report the software, version, company, city and country that developed.

10. Number of replicates was unclear in all assays used. Under the figure, it was mentioned three independent experiments were conducted. But the number of replicates in each experiment was unclear.

Results

1. Figure 2: What is the indicator for ‘A’? Abbreviations were not provided: SnMP, MTS.

2. Line 252 – 254; line 356 - 359 should be in the discussion rather than results.

3. Figure 3D: The * is missing. What is the indication of the black colour bars vs white and grey bars? From the previous figure (Fig. 2), black bars indicate treatment with LPZ.

4. Line 288: ‘Fig. 4D…’ There is no section D in Figure 4.

5. Figure 4 is not cited in the text.

6. Line 331: ‘Nuclear fractions of the lansoprazole-treated cells were incubated in the presence or absence of λPPase…. (Fig. 5A)’. The figure is incorrectly cited in the text. It should be Fig. 4A. Same for Fig. 5B (line 361), it should be Figure 4B.

7. Legend of Figure 5. The description for (B) and (C) has been wrongly presented. Please revise accordingly.

Discussion:

1. To include more limitations of the study.

2. To include the future applications or research direction of this study.

6. PLOS authors have the option to publish the peer review history of their article (what does this mean?). If published, this will include your full peer review and any attached files.

Reviewer #1: No

Reviewer #2: **Yes: **Yow Hui Yin

---

## [Author Response · Author response to Decision Letter 0]

11 May 2023

Answer: We have revised our manuscript according to all of the editors’ and reviewers’ comments. The changes are highlighted in red in the revised manuscript. 

Response to editors

Answer: Thank you for your comments. We have followed the style requirements in the revised version and made some changes. The Keywords and Author Contribution statements were deleted from the manuscript file. Furthermore, Supporting Information text has been added. 

2. Please note that PLOS ONE has specific guidelines on code sharing for submissions in which author-generated code underpins the findings in the manuscript. In these cases, all author-generated code must be made available without restrictions upon publication of the work. 

Answer: We have confirmed that all data underlying the findings described in our manuscript are fully available without restriction and this information has been added to the cover letter. 

“This work was supported in part by a Japan Society for the Promotion of Science 478 (JSPS) KAKENHI grant (grant no. 17K15963) to N. Yamagishi.”

Answer: We have removed the funding information from the Acknowledgements section of the manuscript and this information has been added to the cover letter. 

4. PLOS ONE now requires that authors provide the original uncropped and unadjusted images underlying all blot or gel results reported in a submission’s figures or Supporting Information files.

Answer: We have newly included all the original uncropped western blotting images as Supporting Information. This is also described in in the cover letter.

5. In your Data Availability statement, you have not specified where the minimal data set underlying the results described in your manuscript can be found. PLOS defines a study's minimal data set as the underlying data used to reach the conclusions drawn in the manuscript and any additional data required to replicate the reported study findings in their entirety.

Answer: The following sentence was added to the Data Availability Statement section.

“All data are contained within the manuscript and/or Supporting Information files.”

6. Please amend either the title on the online submission form (via Edit Submission) or the title in the manuscript so that they are identical.

Answer: Thank you. The title is now uniform across all documents and on the online submission form:

“Lansoprazole protects hepatic cells against cisplatin-induced oxidative stress through the p38 MAPK/ARE/Nrf2 pathway.” 

Response to reviewer #1

This manuscript is very interesting and poses lansoprazole as a potential therapeutic target for oxidative stress to the liver. However, the data could be presented in a more clear fashion, and the wording in some areas could be improved. 

Answer: We appreciate your comments and the opportunity to clarify the important points. 

In the revised manuscript, the following points have been corrected:

1. The original title was long and contained a lot of superfluous information; we have revised it to be shorter and more concise.

2. We misquoted the figures in the original manuscript, which were confusing and difficult to understand. In the revised manuscript, we carefully checked and correctly cited the figures.

3. We have revised the wording of sentences throughout the manuscript with the help of a native speaker of English. Moreover, the Methods and Results have been described more precisely. Furthermore, all data referenced in the manuscript and the original uncropped and unadjusted images underlying all blot or gel results have been added to the Supporting Information section. 

4. To strengthen the study, in addition to β-actin, normalization by GAPDH gene expression has been performed in all quantitative RT-PCR experiments. We added the results of quantitative RT-PCR normalized by GAPDH to Figure 1D, Figure 3B, Figure 3D, and Figure 5D. The conclusions of the experiments did not differ with either normalization, suggesting more strongly that lansoprazole induces Nrf2 downstream genes.

5. We revised the limitations of our study. In addition to the two points (ⅱ and ⅳ shown below) listed in the original manuscript, we have added the five limitations to the Discussion section for a total of seven limitations. 

i. The assay was only performed at the third h after lansoprazole administration, so it is possible that there may be peaks in expression or function at other time points. (Lines 465 - 467)

ii. Our study could not demonstrate whether the p38 MAPK directly phosphorylates the Nrf2 in lansoprazole-treated cells. Further studies are required to define the direct association between p38 MAPK and Nrf2. (Lines 483 - 484)

iii. The series of experiments were performed in only one cell line. Lansoprazole exclusively activated p38 MAPK in RL34 cells, other MAPK signaling pathways may be activated in other cell types (e.g., human). (Lines 488 - 489)

iv. This study did not detect a direct modification of the Keap1 protein by lansoprazole. Further studies are required to demonstrate whether the hepatoprotective effect of lansoprazole results from a direct inactivation of Keap1 or from the regulation of the Keap1 inactivation by other upstream factors. (Lines 502 - 506) 

v. One point that was not examined is whether the metabolites of lansoprazole are involved in the antioxidant effect. Lansoprazole is metabolized mainly in the liver to 5'-hydroxylansoprazole and lansoprazole sulfone. If these CYP molecular species are expressed in the RL34 hepatocyte cell line used in this study, the metabolites of lansoprazole may contribute to the antioxidant effect. (Lines 512 - 515)

vi. The lansoprazole concentrations used in our study were higher than those the serum concentrations clinically used. The usual dose of lansoprazole is determined by the level of gastric acid suppression; data on the optimal dose in in vivo studies of Nrf2-mediated hepatoprotective activity is currently insufficient. (Lines 516 - 518)

vii. Importantly, this study only analyzed the expression levels of HO1, NQO1, and GSTA2, which are downstream genes of Nrf2. Lansoprazole has been reported to have anti-inflammatory effects, but the expression levels of inflammatory cytokines were not investigated in this study. (Lines 525 - 528)

6. We have added the following information about future applications and research directions. 

Lines 427 - 429: 

“Lansoprazole is one of the most prescribed drugs worldwide, and consequently has well-established safety and efficacy. Therefore, lansoprazole is expected to expand its clinical application relatively quickly in the future as a treatment for liver diseases.”

　

Lines 536 - 542:

“Oxidative stress is the cause of most liver diseases, and it is a plausible therapeutic target to prevent liver damage [58]. The induction of the antioxidant Nrf2 pathway may protect hepatic cells from oxidative stress and avoid the progression of disease. Various compounds have been postulated to increase Nrf2 activity, although an effective Nrf2 inducer is not currently available to counteract liver diseases. This study extended prior knowledge on the acid-independent effect of lansoprazole and provided novel insights into the mechanisms involved. Thus, lansoprazole may be a potential candidate for the treatment of liver diseases. ”

Response to reviewer #2:

Overview:

Generally, the manuscript presented the novelty of the study in elucidating the potential molecular mechanisms of lansoprazole in cisplatin-induced hepatoxicity in rat hepatic RL34 cells. This manuscript provides an insightful scientific impact on the molecular mechanisms of lansoprazole. The study design and method used were reliable, yet the reporting in methods and results require improvements.

Strengths:

1. This paper presented novel insights on the mechanisms of lansoprazole.

2. The method used was robust and reliable.

Limitations:

1. Poor reporting in methodology and result.

2. Gene expression study with 1 housekeeping gene is less convincing.

Answer: We appreciate your comments and the opportunity to clarify this important point. We have changed as detailed below.

Title: Title is too length with too many unnecessary words. Suggest to revise.

Answer: Thank you for your suggestions. We have shortened the title to the minimum number of characters required as follows:

“Lansoprazole protects hepatic cells against cisplatin-induced oxidative stress through the p38 MAPK/ARE/Nrf2 pathway.” 

Abstract:

1. Lines 14; 28: Kindly avoid the use of first- and second-person pronouns (i.e. we) in the abstract. Please use active voice and third-person narrative throughout the manuscript.

Answer: The first- and second-person pronouns have been revised in the abstract as appropriate, and we have striven to use active voice and third-person narrative in the main part of the revised manuscript. 

2. Line 20: Typo for ‘NAD(P)H:1quinone oxidoreductase-1’? It should be written as ‘NAD(P)H quinone oxidoreductase-1’.

Answer: As you indicated, this was incorrect and has been corrected. 

3. Method was not mentioned in the abstract.

Answer: We added the following information regarding the methodology to lines 17 - 21 of the revised manuscript abstract: 

“Herein, we sought to investigate the molecular mechanism of cytoprotection by lansoprazole. An in vitro experimental model was conducted using cultured rat hepatic cells treated with lansoprazole to analyze the expression levels of Nrf2 and its downstream genes, the activity of Nrf2 using luciferase reporter assays, cisplatin-induced cytotoxicity, and signaling pathways involved in Nrf2 activation.”

Introduction

1. Line 48: ‘The nuclear Nrf2…’

Answer: Please provide further details on what requires correction. We are unsure what you mean here. 

2. Lines 72 - 79: Kindly avoid the use of first- and second-person pronouns (i.e. we) in the abstract. Please use active voice and third-person narrative.

Answer: The first- and second-person pronouns have been corrected into third-person in the abstract as specified.

3. Suggest to include the relationship of nrf-2 with p38 and Erk1 in introduction to provide an better overview on the signaling pathway involved and further justified on the choice of antibodies used in Western blotting.

Answer: The relationship between Nrf-2, p38, and Erk1 is described, and the signaling pathway is outlined in lines 54 - 57 of the Introduction section of the revised manuscript as follows:

“Mitogen-activated protein kinase (MAPK) signaling pathways play important roles in the modulation of ARE-driven gene expression via Nrf2 activation [12-14]. Herein, the capability of lansoprazole to upregulate expression of antioxidant genes via the MAPK/ARE/Nrf2 pathway was investigated using rat hepatic cells.”

Methods

1. Some chemicals or materials only provided with brand name without the city and country of manufacturing, including FBS, DMSO, DMEM, MTS reagent, lansoprazole, RIPA buffer, hygromycin, luciferase assay, anti-rabbit IgG antibody.

Answer: We have added the name of the city and the country of manufacture throughout the Chemicals and Reagents section as specified. 

2. How many replicates run in cell viability assay (MTS)? How many sets of experiment samples per replicate? Please mention in the method.

Answer: Thank you for pointing that out. The cell viability assay (MTS) was performed in three independent experiments, each performed in triplicate. This information has been added to lines 110 - 111 of the method in the revised manuscript as follows: 

“Data for cell viability represent the results of at least three independent experiments, each experiment was performed in triplicate.”

3. Line 106: What 96-well plate reader was used? Please specify the brand of equipment along with the city and country.

Answer: We used a 96-well plate reader manufactured by Corona Electric Co., Ltd. (Ibaraki, Japan). This detail has been added to line 109 of the Methods section in the revised manuscript. 

4. Line 134: CHX abbreviation was used without providing the full term when it was first introduced in the text.

Answer: The full term for CHX (Cycloheximide) is now shown on line 87 of the Methods section in the revised manuscript. 

5. Line 140: Version, city and country of ImageJ?

Answer: The version of Imagej and the names of the city and country have been added to line 146-147 of the revised manuscript as follows: 

“ImageJ ver. 1.53v 21 (U.S. National Institutes of Health, Bethesda, Maryland, USA).”

6. Line 151: The amount of total RNA used to reverse-transcribed into cDNA was not mentioned.

Answer: We used 200 ng of total RNA to create cDNA for all quantitative RT-PCRs. This information has been added to line 159 of the revised manuscript.

7. The experiment setting for gene expression in PCR assay was unclear: volume of reaction mix of PCR, thermal cycling condition, number of replicates, negative control, the validation or optimization of primers, etc.

Answer: Detailed information on PCR assay conditions has been added to lines 159 - 176 of the revised manuscript as follows:

“First strand cDNA was synthesized with the use of the ReverTra Ace qPCR RT Kit (Toyobo, Osaka, Japan) following the manufacturer's instructions in a final volume of 20 μl. The final cDNA fragments were diluted 10-fold before use in RT-PCR. Quantitative RT-PCR analyses were performed using the Brilliant III Ultra-Fast SYBR Green QPCR Master Mix (Agilent Technologies Inc., Tokyo, Japan) and the AriaMX Real-time PCR system (Agilent Technologies Inc.). Each reaction contained 2.5 μL 10-fold diluted cDNA, 200 nM of each primer, and 1 × SYBR Green Master Mix, in a final volume of 10 μL. All reactions were performed in duplicate per cDNA sample. As a control for genomic DNA contamination, total RNA without reverse transcription was tested for each sample per gene. A no-template control was included in each run per gene. The thermal profile of the reaction was 95°C for 5 min activation and denaturation, followed by 40 cycles of 95°C for 10 sec, and 60°C for 10 sec. Finally, a melting curve was generated by increasing temperature starting from 65°C to 95°C to determine the specificity of the reactions. The quantification cycle number (Cq) was determined per reaction with Agilent AriaMx version 2.0. The primers used are listed in Table 1. The relative standard curve method was used to calculate the relative mRNA expression. mRNA expression levels were normalized to those of β-actin and glyceraldehyde-triphosphate dehydrogenase (GAPDH) mRNA. Data represent the results of three independent experiments, and each experiment was performed in triplicate.”

8. Line 157: Only one housekeeping gene was used (beta actin), which is insufficient and at least two housekeeping genes are require for in vitro experiments. Based on MIQE guideline for qPCR, normalization of samples obtained from in vitro experiments can be carried out against a panel (probably two or three) of housekeeping genes whose expression has been shown to be unaffected by experimental conditions. This information must be included in any publication. Refer to MIQE guideline 5th edition published in January 2022. http://www.econferences.de/miqe-talks/

Answer: As a suggestion, in addition to β-actin, normalization by GAPDH gene expression was performed. We have added content to line 175 of the Methods section in the revised manuscript, and the results of quantitative RT-PCR normalized by GAPDH were added to Fig.1D, Fig.3B, Fig.3D, and Fig.5D. There was no difference to the conclusions of the experiment with either normalization. 

9. Line 194: What software was used to run the statistical analyses? Please report the software, version, company, city and country that developed.

Answer: All statistical analyses were performed using JMP version 14.3 statistical software (SAS Institute Inc., Cary, NC, USA). This information has been added to line 214 of the revised manuscript.

10. Number of replicates was unclear in all assays used. Under the figure, it was mentioned three independent experiments were conducted. But the number of replicates in each experiment was unclear.

Answer: All experiments were performed in triplicate, and this detail has been added to each figure legend. 

Results

1. Figure 2: What is the indicator for ‘A’? Abbreviations were not provided: SnMP, MTS.

Answer: Figure 2 is composed of a single figure, so "A" has been removed, and the full terms SnMP and MTS have been added to the legend of Figure 2 (lines 286 - 288 of the revised manuscript).

2. Lines 252 – 254; lines 356 - 359 should be in the discussion rather than results.

Answer: As the reviewer pointed out, we have moved lines 252-254 to lines 437 - 441 in the Discussion section of the revised manuscript as follows. 

“Cisplatin is a commonly utilized chemotherapeutic agent, and confers toxicity to various tissues, including the liver. Growing evidence indicates that the production of ROS induced by cisplatin can trigger oxidative stress, and even cause cell death [16-18, 38, 39]. Increased oxidative stress is one of the main mechanisms involved in cisplatin-induced hepatotoxicity, as indicated by both in vitro and in vivo studies [18].”

Furthermore, we have removed lines 356 – 359 because the same description appears in lines 488- 492 of the Discussion section.

3. Figure 3D: The * is missing. What is the indication of the black colour bars vs white and grey bars? From the previous figure (Fig. 2), black bars indicate treatment with LPZ.

Answer: Thank you for the careful review. We have added the * to the revised Figure 3F (Figure 3D in the original manuscript). There was confusing representation of the color of the bar in Figure 3. In the revised version, we have unified the bars for untreated samples to white and the bars for treated samples to black through the entire figure. 

4. Line 288: ‘Fig. 4D…’ There is no section D in Figure 4.

Answer: Apologies. We made a big mistake about Figure 4. Fig. 4D in the original manuscript has been corrected to Fig. 3F in the revised version. 

5. Figure 4 is not cited in the text.

Answer: Thank you for pointing this out. In the revised manuscript, Figure 4A is now cited on line 340 and Figure 4B is cited on line 344. Figure 5 is cited in the correct position on lines 367 - 371.

6. Line 331: ‘Nuclear fractions of the lansoprazole-treated cells were incubated in the presence or absence of λPPase…. (Fig. 5A)’. The figure is incorrectly cited in the text. It should be Fig. 4A. Same for Fig. 5B (line 361), it should be Figure 4B.

Answer: Thank you for pointing this out. As indicated in the comment #5 above, we have corrected the mistakes. 

7. Legend of Figure 5. The description for (B) and (C) has been wrongly presented. Please revise accordingly.

Answer: We have corrected the legend of Figure 5. 

Discussion:

1. To include more limitations of the study.

Answer: Thank you for your suggestions. In addition to the two points (second and fourth shown below) in the original version of the manuscript, we have added five limitations of this study to the Discussion section, for a total seven limitation, which are listed below. 

Lines 465- 467: 

“One limitation of this study, based on previous reports [32], is that sampling was conducted at 3 h after lansoprazole treatment, and stronger antioxidant activity could be observed if tested with a longer time course.”

Lines 483- 485: 

“However, as a second limitation, our study could not demonstrate whether the p38 MAPK directly phosphorylates the Nrf2 in lansoprazole-treated cells. Further studies are required to define the direct association between p38 MAPK and Nrf2.”

Lines 488- 491: 

“A third limitation is that the experiments were performed in only one cell line. Although lansoprazole caused exclusive p38 MAPK activation in RL34 cells, it is possible that other MAPK signaling pathways are activated in other cell types (e.g., human).”

Lines 502- 506:

“However, as a fourth limitation, this study did not detect a direct modification of the Keap1 protein by lansoprazole. Further studies are required to demonstrate whether the hepatoprotective effect of lansoprazole results from a direct inactivation of Keap1 or from the regulation of the Keap1 inactivation by other upstream factors.”　

Lines 512- 515: 

“As a fifth limitation, the present study could not determine whether the cytoprotective effect of lansoprazole was caused by acid-induced activation or metabolism by CYP emzymes, and this requires future investigation.”

Lines 516 - 518:

“The concentration of 100 μM lansorazole employed in this study is much higher than the maximum serum concentration in healthy adults given a single oral administration of 30 mg lansoprazole, which corresponds a sixth limitation.”

Lines 525 - 528:

“As a seventh limitation, lansoprazole is reported to have anti-inflammatory effects [19, 20], but the expression levels of inflammatory cytokines were not investigated in this study. Therefore, subsequent studies should focus on inflammatory cytokines and investigate gene expression levels, focusing on those downstream of Nrf2.”

2. To include the future applications or research direction of this study.

Answer: Thank you for your suggestion. Based on this study, we have added the following to the Discussion section for future applications and research directions for lansoprazole. 

Lines 427- 429:

“Lansoprazole is one of the most prescribed drugs worldwide, and consequently has well-established safety and efficacy. Therefore, lansoprazole is expected to expand its clinical application relatively quickly in the future as a treatment for liver diseases.”　

Lines 536- 542: 

“Oxidative stress is the cause of most liver diseases, and it is a plausible therapeutic target to prevent liver damage [58]. The induction of the antioxidant Nrf2 pathway may protect hepatic cells from oxidative stress and avoid the progression of disease. Various compounds have been postulated to increase Nrf2 activity, although an effective Nrf2 inducer is not currently available to counteract liver diseases. This study extended prior knowledge on the acid-independent effect of lansoprazole and provided novel insights into the mechanisms involved. Thus, lansoprazole may be a potential candidate for the treatment of liver diseases.”

---

## [Editor Report · Decision Letter 1]

17 May 2023

PONE-D-22-35612R1Lansoprazole protects hepatic cells against cisplatin-induced oxidative stress through the p38 MAPK/ARE/Nrf2 pathwayPLOS ONE

Dear Dr. Yamagishi,

Thank you for submitting your manuscript to PLOS ONE. After careful consideration, we feel that it has merit but does not fully meet PLOS ONE’s publication criteria as it currently stands. Therefore, we invite you to submit a revised version of the manuscript that addresses the points raised during the review process.

We look forward to receiving your revised manuscript.

Kind regards,

Wei Hsum Yap

Academic Editor

PLOS ONE

Journal Requirements:

Additional Editor Comments:

Dear authors, please include the following revisions:

1. There is no explanation for relationship between Nrf2, p38 and ERK1 in Introduction section Line 53-57. Please include justification on why these kinases are included in the current study.

2. Please include the final concentration or quantity of cDNA that was used in the reaction (Line 161-162)

3. Please include CHX abbreviations in Fig 1 caption - each diagram is independent, and should include info for abbreviated text within the image

4. Please change "single administration of" (Line 330) to cisplatin treatment group

5. Line 336-337 the sentence is ambiguous because Fig 1A is showing Western blot analysis instead of SDS-PAGE. Suggest rephrasing or removing the sentence to provide a better flow of explanation.

---

## [Author Response · Author response to Decision Letter 1]

8 Jun 2023

Answer: We have carefully reviewed the references list and properly revised it, and we have ensured its correctness and completeness. 

Additional Editor Comments:

Dear authors, please include the following revisions:

1. There is no explanation for relationship between Nrf2, p38 and ERK1 in Introduction section Line 53-57. Please include justification on why these kinases are included in the current study. 

Answer: We appreciate your comments and the opportunity to clarify these important points. The relationship between Nrf2 and MAPKs is described in lines 56 - 74 of the revised manuscript as follows: 

“Exposure of cells to Nrf2 inducers, like oxidative stress stimuli, activates Nrf2/ARE-mediated response through different intracellular signaling pathways. Mitogen-activated protein kinase (MAPK) pathways are known to regulate Nrf2/ARE-driven gene expression [12-14]. MAPKs are serine/threonine protein kinase that play a central role in the signaling cascade regulating cellular processes, such as cell proliferation, differentiation, and apoptosis. Three major MAPK subfamilies have been extensively studied: extracellular signal-regulated kinases (ERKs), c-Jun N-terminal kinases (JNKs), and p38 MAPK. Each kinase establishes, in principle, parallel and independent signaling pathways. Depending on the cellular and stimulatory context, there is often significant cross-talk between each of the kinases because they can respond to common upstream activators and phosphorylate common downstream targets. Alam et al. reported that cadmium-induced HO1 gene expression requires the sequential activation of the p38 MAPK pathway and Nrf2 in human breast cancer MCF-7 cells [12]. In contrast, in human hepatoblastoma HepG2 cells, JNK activated the induction of Nrf2/ARE-mediated gene expression by the overexpression of common upstream kinases, whereas p38 MAPK showed the opposite effects [13]. Additionally, exposure of HepG2 cells to the chemical pyrrolidine dithiocarbamate, the activation of both ERK1/2 and p38 MAPK are required for induction of the ARE-mediated γ-glutamylcysteine synthetase subunit genes [14]. Based on these observations, the capability of lansoprazole to upregulate expression of antioxidant genes via the MAPK/ARE/Nrf2 pathway was investigated using rat hepatic cells.”

2. Please include the final concentration or quantity of cDNA that was used in the reaction (Line 161-162)

Answer: We used 2.5 μl per well (2.5 ng/reaction) of 10-fold diluted cDNA solution for all quantitative RT-PCR. This information has been added to lines 180 and 183 of the revised manuscript.

3. Please include CHX abbreviations in Fig 1 caption - each diagram is independent, and should include info for abbreviated text within the image

Answer: We have added the abbreviation for CHX to Fig. 1 caption (Lines 283 - 284 of the revised manuscript). 

4. Please change "single administration of" (Line 330) to cisplatin treatment group

Answer: We have changed this to “cisplatin treatment group” (Line 352 of the revised manuscript). 

5. Line 336-337 the sentence is ambiguous because Fig 1A is showing Western blot analysis instead of SDS-PAGE. Suggest rephrasing or removing the sentence to provide a better flow of explanation.

Answer: Thank you for pointing this out. In the revised manuscript, lines 357 - 359 have been changed as follows:

“The Nrf2 band was found to be upshifted in the nuclear fraction of the lansoprazole-treated cells (Fig. 1A) indicating that Nrf2 translocated to the nucleus underwent posttranscriptional modifications.”

---

## [Editor Report · Decision Letter 2]

13 Jun 2023

Lansoprazole protects hepatic cells against cisplatin-induced oxidative stress through the p38 MAPK/ARE/Nrf2 pathway

PONE-D-22-35612R2

Dear Dr. Yamagishi,

We’re pleased to inform you that your manuscript has been judged scientifically suitable for publication and will be formally accepted for publication once it meets all outstanding technical requirements.

Kind regards,

Wei Hsum Yap

Academic Editor

PLOS ONE
---

## [Editor Report · Acceptance letter]

22 Jun 2023

PONE-D-22-35612R2 

Lansoprazole protects hepatic cells against cisplatin-induced oxidative stress through the p38 MAPK/ARE/Nrf2 pathway 

Dear Dr. Yamagishi:

I'm pleased to inform you that your manuscript has been deemed suitable for publication in PLOS ONE. Congratulations! Your manuscript is now with our production department. 

Kind regards, 

on behalf of

Dr. Wei Hsum Yap 

Academic Editor

PLOS ONE